# Finally Freed—*Cannabis* in South Africa: A Review Contextualised within Global History, Diversity, and Chemical Profiles

**DOI:** 10.3390/plants13192695

**Published:** 2024-09-26

**Authors:** Valencia V. Ndlangamandla, Adeola Salawu-Rotimi, Vuyiswa S. Bushula-Njah, Nompilo L. Hlongwane, Gugu F. Sibandze, Fikisiwe C. Gebashe, Nokuthula P. Mchunu

**Affiliations:** 1School of Life Sciences, College of Agriculture, Engineering and Sciences, University of KwaZulu-Natal, Westville Campus, Private Bag X 54001, Durban 4000, South Africa; ndlangamandlav@arc.agric.za (V.V.N.); np.mchunu@nrf.ac.za (N.P.M.); 2Agricultural Research Council-Biotechnology Platform Onderstepoort Veterinary Research, Private Bag X 5, Onderstepoort 0110, South Africa; talk2adey100@gmail.com (A.S.-R.); vuyiswa@ksu.edu (V.S.B.-N.); hlongwanen@arc.agric.za (N.L.H.); 3Eswatini Institute for Research in Traditional Medicine, Medicinal and Indigenous Food Plants, University of Eswatini, Private Bag 4, Kwaluseni M201, Eswatini; gsibandze@uniswa.sz; 4National Research Foundation, Meiring Naude, Pretoria 0001, South Africa

**Keywords:** *Cannabis*, legalisation, Δ9-tetrahydrocannabinol/Δ^9^-THC, cannabidiol/CBD, genomics

## Abstract

*Cannabis sativa* L. is a monotypic genus belonging to the family Cannabaceae. It is one of the oldest species cultivated by humans, believed to have originated in Central Asia. In pivotal judgements in 2016 and 2018, the South African Constitutional Court legalised the use of *Cannabis* within the country for medicinal and recreational purposes, respectively. These decrees opened opportunities for in-depth research where previously there had been varying sentiments for research to be conducted on the plant. This review seeks to examine the history, genetic diversity, and chemical profile of *Cannabis*. The cultivation of *Cannabis* by indigenous people of southern Africa dates back to the eighteenth century. Indigenous rural communities have been supporting their livelihoods through *Cannabis* farming even before its legalisation. However, there are limited studies on the plant’s diversity, both morphologically and genetically, and its chemical composition. Also, there is a lack of proper documentation of *Cannabis* varieties in southern Africa. Currently, the National Centre for Biotechnology Information (NCBI) has 15 genome assemblies of *Cannabis* obtained from hemp and drug cultivars; however, none of these are representatives of African samples. More studies are needed to explore the species’ knowledge gaps on genetic diversity and chemical profiles to develop the *Cannabis* sector in southern Africa.

## 1. Introduction

In September 2018, the South African Constitutional Court ruled in a landmark case that it will be legal for an adult to use or possess *Cannabis* within South Africa [1]. This ruling spurred research on *Cannabis*, not only for medicinal and industrial purposes, but also for recreational purposes [2]. *Cannabis sativa* L. is one of the oldest cultivated species hypothesised to have originated in Central Asia and belongs to the family Cannabaceae [3,4,5]. There is still an ongoing debate between scientists regarding the taxonomy of the plant [3]. *C. sativa* L. is known as a monotypic genus of the Cannabaceae family [3,6]. However, other scientists have divided *Cannabis* into three distinct species, namely, *C. sativa* L., *Cannabis indica* Lam., and *Cannabis ruderalis* Janisch. [7]. But it is now generally accepted that the genus is only represented by one polymorphic species, *C. sativa* L. [3,5,6]. In this review, the plant will be referred to by its scientific genus name, *Cannabis*. The initial discovery and cultivation of *Cannabis* by humans can be traced back to a period ranging from 6000 to 12,000 years in Central Asia, particularly in Mongolia and southern Siberia [5,8,9]. Arab traders, while traversing India, are believed to have introduced the plant to Africa during the 1300s [10,11]. According to Warf [8] and Duvall [12], *Cannabis* made its way to Southern Africa between the 1500s and 1700s as a recreational crop through trade and European/Portuguese travel and settlement. South Africa and its neighbouring country, Eswatini, have favourable climates for growing *Cannabis* [13] and have been cultivating it for at least six centuries [14]. Due to the varying climatic and agroecological conditions between these two countries, there may be *Cannabis* strains that are climatically adapted to these regions and have a potentially rich genetic reservoir that has never been explored before. Although national and global studies on *Cannabis* are still in their infancy, exploration of this species’ genetic diversity and chemical composition is vital to sustain and develop the *Cannabis* sector in South Africa. Within the South African *Cannabis* varieties, not much is known about markers associated with important traits such as cannabinoid expression, fibre, and resin production of the local *Cannabis* populations. Therefore, reviewing the origins, history, and comprehension of genetic diversity and chemical (cannabinoid) profiling of *Cannabis* are the primary objectives of this review.

## 2. Origin and History of *Cannabis*

The origin and first discovery of *Cannabis* was in Central Asia, specifically in Mongolia and southern Siberia. *Cannabis* spread from Mongolia to the Middle East and Eastern Europe between 3000 to 6000 years ago in Siberian regions [8]. The discovery of *Cannabis* goes as far back as 12,000 years, and it is among the oldest cultivated crops known to man (Figure 1) [4,10,12]. In Central Asia, *Cannabis* (hemp) was widely for manufacturing rope, clothing, sails, and bowstrings and as a psychoactive agent [8,10]. The use of the crop gradually increased in Central Asia and spread westward, reaching India about 3000 years ago (Figure 1) [4,10,12]. The general uses of the plant in India included medicinal purposes (analgesic, anticonvulsant, antibiotic, and anti-inflammatory), and it was also smoked during devotional services by Hindus [10]. In the 1300s, Arab traders brought *Cannabis* from India to Africa (Figure 1) [4,10,12]. The plant entered Africa across the Red Sea via Egypt and Ethiopia, where it was used mostly by Muslims for traditional rituals and smoking [8]. According to Hand et al. [15], *Cannabis* in Africa was also used for medicinal purposes to treat malaria and snake bites [8].

In the 16th century, *Cannabis* was introduced to South America (Figure 1) [4,10,12], via the slave trade, where Africans were shipped along with seeds from Angola to Brazil [10]. In Brazil, it was used by African communities for religious rituals and magical and medicinal purposes to treat malaria, fever, asthma, and the infection of the intestines that causes diarrhoea and painful stomach cramps [10,12]. From Brazil, the plant spread to Colombia and travelled to North America via Mexico in the early 20th century (Figure 1) [4,10,12], where it was used for recreational purposes by individuals in the low-economic class [8,10]. According to Hand et al. [10], researchers found that the first introduction of *Cannabis* to Western medicine was documented in Europe in 1839 by an Irish physician, William O’Shaughnessy. William O’Shaughnessy studied Indian hemp, also known as Gunjan, for analgesic and sedative properties [10].

Following its initial introduction into Africa, *Cannabis* was initially grown on the continent as a recreational crop. Sub-Saharan Africa is regarded as the origin of smoking pipes [13]. However, over the years, African societies began using the plant in multiple ways [12]. For example, farmers in North Africa grew hemp in the early 1900s to make hemp cloth [12]. However, in the 1790s, Mozambican farmers planted *Cannabis* primarily for its inhalable inflorescences and leaves for medicinal purposes [12,13]. In South Africa, as discussed before, *Cannabis* was introduced during the 1500s and 1700s (Figure 1) [4,10,12] by the Portuguese travelling in ships [8,12]. Initially, the plant was cultivated on a small scale for smoking, rituals, and customs [16]. However, once the plant was made legal in September 2018, it was increasingly used for export, research, medicinal and recreational purposes [17].

### 2.1. Decriminalisation and Legalisation of Cannabis

Decriminalisation is the act of removing criminal penalties against an act, article, or behaviour [18]. Decriminalisation of *Cannabis* means that although *Cannabis* remains illegal, possession of less than 1 g of dried *Cannabis* or five grams of fresh *Cannabis* would not face legal action in South Africa [15]. Legalisation of *Cannabis* is the removal of all legal prohibitions such as cultivating, smoking, or selling against the plant [18]. Decriminalisation and legalisation of *Cannabis* for recreational and medicinal purposes differ widely by country and region, but the plant’s cultivation/selling/smoking remains largely prohibited globally [15]. This means that *Cannabis* will be available to the adult general population to buy and use, similar to alcohol and tobacco [15,18]. However, it is still illegal to sell *Cannabis* for recreational use, which seems to imply that all users are/should be growing their plants, which in practicality is not true.

#### 2.1.1. Decriminalisation of the Use of Cannabis in South Africa

After Lesotho, South Africa became the second country in Africa to legalise *Cannabis* for medicinal use [16]. However, Lubaale and Mavundla [19] report that South Africa is the first nation to legalize the plant for private use and consumption. According to Singh [20], South Africa began reviewing the prohibitions on growing and possession of *Cannabis* in 2000. The South African Constitutional Court finally decriminalised *Cannabis* use for medical (2016) and recreational (2018) purposes [1], allowing this plant species to be legally explored for medicinal, industrial, and recreational purposes [21]. The legalisation of *Cannabis* has raised concerns about the plant’s social and health impacts, use in children, illegal cultivation, increase in the number of users, etc. [1]. This led to the drafting of a bill by the Minister of Justice and Correctional Services, Justice Zondo, in September 2020 [1], to use *Cannabis* for private purposes.

#### 2.1.2. The Summary of the South African Bill of Cannabis for Private Purposes

The Constitutional Court ruled in 2018 that *Cannabis* usage is acceptable for both personal and medicinal purposes [1,21]. However, it remains illegal to buy and sell C*annabis*, as well as to use it outside of one’s own house. The draft bill aimed at protecting an adult’s right to privacy if they own *Cannabis* growing equipment, grow a certain number of *Cannabis* plants, or use *Cannabis* for private purposes [2,21]. The draft bill will also provide the clearance of new criminal records for those convicted of *Cannabis* possession or use and eliminate or modify provisions of some *Cannabis* prohibition laws [1]. The summarised rules are as follows:An adult person may use or be in possession of *Cannabis* in private for personal consumption. The personal limit for possession outside the home has been set to a maximum of 1 g of dried flower, 6 g in private space, and 12 g for households with two or more adults.Smoking *Cannabis* in public or in the presence of children or the presence of a non-consenting adult is not permitted.The cultivation of *Cannabis* by an adult in a private place for personal consumption is no longer a criminal offence. However, growing limits have been set to a maximum of four plants per single person or eight plants for two or more adults sharing a household.Large or small-scale farmers need permits for growing and selling *Cannabis*.Offences could lead to a maximum of 6 to 15 years of prison sentences.

### 2.2. General Uses of Cannabis in South Africa and Eswatini

*Cannabis* is famous for being a multipurpose plant [4,12]; for example, the general use of *Cannabis* in South Africa includes burning/smoking during spiritual/cultural rituals and recreational and medicinal purposes [2]. Its use depends on the specific variety of the plant [22]. The varieties are classified as marijuana in various parts of the world and dagga in South Africa (drug type) [12], and hemp or industrial hemp (fibre type) [23,24].

For many years, there has been a lack of understanding regarding the proper differentiation between hemp and marijuana [25]. The division of *Cannabis* strains into hemp and marijuana is due to the different cannabinoids contained [26]. Cannabinoids refer to one or more of the approximately 200 compounds or phytochemicals found in *Cannabis*’s trichomes [27,28].

Researchers from various nations present varying arguments regarding the percentage of Δ9-tetrahydrocannabinol (Δ^9^-THC) and Cannabidiol (CBD) in hemp and drug type [5,22,27]. Hemp should contain less than 0.2–1% Δ^9^-THC, depending on national laws of different countries, whereas dry inflorescences of drug type could contain up to 20–30% Δ^9^-THC [27]. Many researchers from North America and Europe have observed that plants classified as fibre contain less than 0.3% Δ^9^-THC, while plants classified as a drug contain more than 0.3% Δ^9^-THC [5,22,29].

#### 2.2.1. Cannabis for Medicinal Application

*Cannabis,* such as dagga or marijuana, has a long history of use for recreational and medicinal purposes [28]. Current *Cannabis* research is more focused on its properties for pain relief, nausea, vomiting, appetite stimulation and anti-inflammatory activity associated with cancer, HIV/AIDS, and COVID-19 (Table 1) [25,30]. The first study to ever outline the potential of *Cannabis* as a medical plant was previously released in 1843 by O’Shaughnessy [31], where it was explored for the treatment of bacterial diseases such as cholera and tetanus. Since 2001, researchers in Canada have been conducting research on the use of *Cannabis* for medicinal purposes [32]. The studies have focused on the efficacy of cannabinoid medications as potential treatments for conditions like multiple sclerosis, mental illnesses, epilepsy, inflammatory diseases, cancer, obesity, glaucoma, and neurodegenerative disorders [33,34].

According to recent research from Italy, Canada, Australia, the United Kingdom, and the United States of America, medicinal *Cannabis* may prevent chemotherapy-induced nausea and vomiting, increase appetite, lessen discomfort, reduce inflammation, and increase cancer cell death and proliferation [25,39]. The cannabinoid Δ^9^-THC has neuroprotective, antispasmodic, and anti-inflammatory actions, which play a huge role in cancer by inhibiting the cell growth of many tumours [36]. One of the world-first clinical studies to examine the first *Cannabis*-based drug, Sativex^®^, in treating massive brain tumours in patients was conducted by researchers at the University of Leeds and the Cancer Research UK Clinical Trials Unit at the University of Birmingham, United Kingdom in 2023 [40]. Part of the goals of their ongoing research is to ascertain whether using Sativex^®^ and chemotherapy together will assist patients with recurrent glioblastoma to survive longer [40].

In South Africa, *Cannabis* has been mostly used in traditional medicine for the treatment of hypertension, diabetes, diarrhoea, pains, epilepsy, and bronchial diseases (such as coughing and asthma) (Table 2) [41,42,43,44]. There are very few studies in South Africa on the medical application of *Cannabis,* which is not surprising due to its previous legal status. Dronabinol was the only synthetic medicinal *Cannabis* product that the South African Health Products Regulatory Authority (SAHPRA) has authorized thus far [17]. Medicinal *Cannabis* (dronabinol) was used to treat Lennox–Gastaut syndrome and Dravet syndrome, two drug-resistant childhood forms of epilepsy [17].

#### 2.2.2. Cannabis for Agricultural, Textile and Industrial Applications

Hemp usually refers to *Cannabis* strains that are mostly grown for agricultural purposes (seeds, fibres, and CBD-dominant inflorescences) [24]. Singh et al. [23] reported that hemp is the oldest source of fibre in Central Asia for manufacturing ropes and clothes. More recently, it has been grown for industrial purposes to obtain oil, fibre, and food [24]. Hemp growers generally plant the male plants to produce hemp fibre and seeds. To encourage more fibre production (taller stalk growth and less branching), growers prevent flowering by pruning and pinning, while those who grow the plant for CBD will promote flower growth [23,45]. Hemp is further grown for food (CBD drinks and edibles) and nutritional purposes as well as in cosmetics (CBD skin, hair care, and toothpaste) [23].

### 2.3. Cannabis as an Economic Crop in Rural Communities of South Africa

According to recent press sources, more than 900,000 small-scale farmers in the KwaZulu-Natal and the Eastern Cape provinces have been cultivating *Cannabis* for more than 200 years [46], with Mpondoland in the Eastern Cape being the largest *Cannabis*-growing region in South Africa [2]. Rural communities in these provinces have been growing the plant even when it was still illegal to grow *Cannabis* in South Africa [47]. In addition to these two provinces, Mpumalanga and Limpopo are two other important traditional-growing regions in South Africa [2]. People in these regions generate their livelihoods through a mixture of crop and livestock farming [47]; however, most depend entirely on *Cannabis* for income [48].

Most small-scale farmers in rural South African settlements are women who are frequently the family’s sole breadwinners [49]. According to Ablin et al. [38], the farmers depend on *Cannabis* as a source of income because there are no jobs to generate income. *Cannabis* is key to the livelihoods of many villagers in rural communities [47]. Villagers grow the plant to sell locally or to big cities and in one interview, one of the small-scale growers mentioned that in a good month, revenue generated from selling *Cannabis* can be about ZAR 2800.00 (approximately USD 151.47) (personal interviews) [50]. Also, growers who transport their *Cannabis* to cities experience greater financial rewards [46]. For instance, in a 2003 case study by Kepe [47], individuals who sold *Cannabis* in cities like Durban reported a monthly profit of ZAR 20,000 (approximately USD 1000+). However, trading *Cannabis* across provinces or cities means farmers take a risk of getting arrested or paying fines or bribes to law enforcement to avoid arrests or fines. Furthermore, transportation of the crop to different destinations was an unwelcome expenditure [47].

While supporting the legalisation of the plant, in recent reports, South African *Cannabis* growers feel left behind in the legalisation plans and the *Cannabis* bill [46]. In a report by Clark and Hendricks [49], *Cannabis* growers were concerned that the legalisation of the crop opened the *Cannabis* market to large businesses and commercial farmers but left out rural *Cannabis* farmers. The growers cite the high cost associated with acquiring *Cannabis* production permits as the main hindrance to market entry. Currently, the only way to engage in legal *Cannabis* commerce in South Africa is to secure a medical *Cannabis* production license from the South African Health Products Regulatory Authority (SAHPRA) [13]. The cost of preparing for a licence application and the application fee is very expensive for small-scale rural farmers [48]; the prescribed license fee is ZAR 902.00 (approximately USD 48.80) for *Cannabis* (hemp) and ZAR 23,980.00 (approximately USD 1300.00) for a *Cannabis* (medicinal) application, with an inspection fee of ZAR 714 (approximately USD 38.62) per hour and a hardcopy license collection fee of ZAR 3180 (approximately USD 172.03), the estimated total cost for hemp is ZAR 4796.00 (approximately USD 259.45), and for medicinal *Cannabis* is ZAR 27,874.00 (approximately USD 1507.93) [50]. Furthermore, aside from being expensive, the process of preparing for a licence application could be difficult for most farmers in rural settlements. A professional business cover letter and the SAHPRA licence (Building A, Loftus Park, 402 Kirkness street, Arcadia, Pretoria, 0001, South Africa) must be downloaded from the SAPHRA website (https://www.sahpra.org.za, accessed on 22 July 2024). For the most part, farmers are at a disadvantage because they have limited or no access to information and communication technologies. Rural *Cannabis* farmers will become marginalised as a result. The government ought to enact policies or legislation that will be citizen-friendly and enable rural farmers to compete with the *Cannabis* global market. This section highlights the significant potential that *Cannabis* holds to revitalize rural communities through the creation of jobs, local economic growth, enhanced access to healthcare, and promotion of sustainable agricultural practices. Rural areas have the potential to achieve long-term growth and development by utilising these advantages through prudent management and inclusive policies.

### 2.4. Cannabis Landraces in Southern Africa

The Kingdom of Eswatini previously known as Swaziland is a prominent producer of *Cannabis* in southern Africa where over 70% of small-scale farmers growing *Cannabis* (famous strain nicknamed Swazi Gold) sell and export it to other countries like South Africa [16]. Similarly, South Africa has a favourable climate for growing *Cannabis* [14], producing 2500 tonnes of *Cannabis* per year [13]. The two countries share regions with high mountains, which offer plentiful water sources and fertile soils ideal for farming *Cannabis* [16]. In South Africa, the plant is referred to by various names depending on the language spoken, such as Dagga (Afrikaans), Umya or Intsangu (Xhosa), Insangu (Zulu/Swati), Matakwane, Matokwane, Matekwane, or Mmoana (Sotho) [2,47].

The major growing regions of *Cannabis* in South Africa include the Eastern Cape (EC), Mpumalanga (near Eswatini), KwaZulu-Natal (KZN), and Limpopo province [2], with the EC and KZN producing the bulk of *Cannabis* outputs [47]. The EC, particularly Mpondoland, has become world-famous for its *Cannabis* strain, nicknamed South African’s Mpondo Gold and previously known as Transkei Gold [2]. This strain has gained its fame as the Swazi Gold from the Kingdom of Eswatini and Durban Poison from the city of Durban, South Africa [2]. However, there has been no study on *Cannabis* landraces in southern Africa. The names of the landraces are based on the language spoken and the region where the plant comes from [2]. This suggests that different names are likely applied to the same *Cannabis* landraces and vice versa [51]. The lack of accurate information and nomenclature regarding landraces in South Africa results in incorrect species identification. These landraces are known to have distinct genetic profiles and are vital sources of genetic diversity in breeding programs [52]. To comprehend the diversity of *Cannabis* in the country, more emphasis needs to be paid to the identification and description of the local landraces. This is crucial for knowing the exact species varieties’ names and understanding the morphological and genetic traits that aid future plant breeding programs to improve *Cannabis* yield, growth, and cannabinoid production such as Δ^9^-THC and CBD [28,53].

## 3. Cannabis Diversity

### 3.1. Botanical Taxonomy of Cannabis

There is confusion and current debate among botanists and taxonomists regarding the taxonomy of *Cannabis* [3]. *Cannabis sativa* L. is known as a monotypic genus of the Cannabaceae family [3,4,5,6]. As a result of continuous taxonomic research within the genus *C. sativa* L., a considerable number of names have been proposed at the rank of species or intraspecific taxa [54]. The reason for such an excess of names is the great morphological habitat-dependent variability of *Cannabis,* which the botanists tried to assign a taxonomic value. All these historical names have today lost their taxonomic value in favour of a single species, *C. sativa* L. [6]. Only a few authors are inclined to recognise three taxa*: C. sativa* L., *C. indica* Lam., and *C. ruderalis* Janisch. [6,7,55]. In formal botanical nomenclature, *C. sativa* contains three subspecies, namely *C. sativa* subsp. *sativa* L.*, C. sativa* subsp. *indica* Lam.*,* and *C. sativa* subsp. *ruderalis* Janisch. [6,22]. The two subspecies *C. sativa* subsp. *sativa* and *C. sativa* subsp. *indica* each have two varieties, which include *C. sativa* subsp. *sativa* var. *sativa*, and *C. sativa* subsp. *sativa* var. *spontanea* as well as *C. sativa* subsp. *indica* var. *indica* and *C. sativa* subsp. *indica* var. *kafiristanica* [6,54,56].

### 3.2. Cannabis Morphological Diversity

*Cannabis* is a dioecious wind-pollinated flowering plant [28], having male and female flowers developed in separate individuals (Figure 2) [53]. Furthermore, the plant can become a hermaphrodite (Figure 2), meaning that it will produce both male and female flowers on the same plant [3]. This is rare and is normally a consequence of environmental factors [3,57,58]. The plant has different varieties with a strong aroma; male flowers are found in axillary cymose panicles, and female flowers are clustered at the apices [53]. Male plants are green and usually taller, and female plants are dark green and are more branched than male plants [55,58].

The female flower has a single ovule, which is enriched with trichomes [57,59]. There are two types of trichomes present in *Cannabis*: glandular and non-glandular [59]. The glandular trichomes are the main structures for the synthesis and storage of cannabinoids [28]. Since the floral biomass of male and hermaphrodite plants is lower, the yield of phytocannabinoid compounds in these plants is also low [59]. *Cannabis* plants can reach a height of 3 m or less with palmate leaves carrying 5–7 leaflets, depending on the variety and growing conditions [57]. The stem is made up of the epidermis’ thin and coarse outer layer, as well as primary and secondary layers that provide better fibre quality [57].

The well-documented strains are *C. sativa* subsp. *sativa* and *C. sativa* subsp. *indica* [3,6]. *C. sativa* subsp. *sativa* strains are distinguished by their tall height, widely spaced branches, and long and thin leaves (Figure 3) [55]. In contrast, *C. sativa* subsp. *indica* strains are characterized as shorter and highly branched plants with broader leaves (Figure 3) [3,55]. The subspecies, *C. sativa* subsp. *ruderalis* is not as popular; the plant is an heirloom with short stature, less branching, and small thick leaves (Figure 3) [6,55].

The diversity of morphological traits plays a significant role in cannabinoid content, physiology, quality and yield potential of the *Cannabis* plant [60]. Thus, studies on the morphological diversity of the plant can provide insights into the plant’s cultivation, medical application and breeding processes. Hurgobin et al. [29] reported that *Cannabis* plants with shorter, thinner stems, more branches and a greater floral tissue density are used for medical and recreational purposes. Breeding programs aimed at high-quality *Cannabis* varieties with desired traits can be influenced by precisely recognising and describing these morphological traits and can help improve the *Cannabis* industry [54].

### 3.3. Cannabis Genetic Diversity

The genome of *Cannabis* is diploid (2*n* = 20) composed of nine autosomes and a pair of sex chromosomes (X and Y) [29,61,62]. The estimated haploid genome size for female plants is 818 megabase pairs (Mbp) and 843 for male plants [28]. According to Braich et al. [63], the diploid genome size for the *Cannabis* plant when using flow cytometry is 1636 ± 7.2 Mbp and 1683 ± 13.9 Mbp for female and male plants, respectively.

In 2011, the first draft genome for a *Cannabis* (marijuana) strain was published in Canada [64]. This Canadian draft genome (marijuana strain; Purple Kush (PK) had been sequenced using Illumina shotgun and mate-pair sequencing and produced 786 Mbp in 136,290 unmapped scaffolds, with 534 Mbp called in 363,760 contigs and 252 Mbp uncalled (N) in 228,430 gaps [28,63], and genome annotation of 2781 predicted genes [61,63]. This draft genome opened doors in genomic studies of *Cannabis*, as it was first used as a reference genome [5]. According to Lapierre et al. [5], there may be a significant genetic difference between hemp- and drug-type *Cannabis*. The authors reviewed the comparison of the Purple Kush with the Canadian Finola, a hemp-type *Cannabis*, the marijuana Purple Kush has a larger genome size when compared to the estimated hemp Finola (Appendix A) [61,63].

Currently, the National Centre for Biotechnology Information (NCBI) has 15 genome assemblies (chromosome-resolved, scaffold, or contig level) that were created using both short and long read sequencing [5,29]; their statistics are summarised in (Appendix A). There were significant variations in the assembled genome size across the reported genomes (Appendix A).

The United States CBDRx, commonly referred to as cs10, is the most comprehensive *Cannabis* reference genome sequenced with Oxford Nanopore technology [61], with 854 Mbp in 773 scaffolds (Appendix A) [61]. In the updated version of this assembly (GenBank no. GCA_900626175.2), Grassa et al. [61] predicted the presence of 25,302 protein genes in the cs10 assembly and the BUSCO analysis identified 97% out of 425 BUSCOS as complete. Compared to other *Cannabis* genome assemblies, the cs10 genome assembly has a larger N50 of 91.9 MB, indicating good genome quality [61,65].

In early 2020, the International *Cannabis* Genomics Research Consortium recommended that the assembly cs10 be used as the reference for *Cannabis* genomics and the current version of this assembly has been updated with the chromosomes renumbered [5,29]. As the legalisation of medicinal and recreational *Cannabis* gains ground in South Africa and across the globe [2,20], more doors have been opened for research to foster and grow the *Cannabi*s industry in the country. To date, no studies have been published that compare the South African or African *Cannabis* genome against the 15 published genomes. Moreover, there are no published South African or African representative genomes by the NCBI. The field of *Cannabis* genomics is evolving rapidly, and studies on African or South African genomes that could be used as a reference in future population genetic studies of *Cannabis* are needed. The African or South African *Cannabis* genome will provide a standard to compare our genomes and will also be used as a reference in future population genetic studies here and abroad. This is essential for identifying and understanding the genetic variation of *Cannabis* in South Africa with the use of high-throughput methods such as genotyping-by-sequencing (GBS) and also single-nucleotide polymorphisms (SNPs) [29]. Furthermore, it could be used in breeding programs to perform marker-assisted selection and discover genes underlying phenotypic variation in *Cannabis* [53].

We believe that the use of long-read sequencing technologies could play a significant role in creating chromosome-scale whole genome sequence assemblies for South African *Cannabis*. This will provide information about the genome-wide sequence, which will help the *Cannabis* medicine industry’s ability to select and edit genomes as well as analyse pan- genome sequences [63].

### 3.4. Population Structure of Cannabis Using Single Nucleotide Polymorphisms (SNPs)

Single nucleotide polymorphisms (SNPs) have become increasingly significant in genomic research as co-dominant markers of the genome identified through high-throughput genotyping [66]. The term SNPs refers to polymorphisms brought on by point mutations that result in different alleles with variation bases Adeline (A), Guanine (G), Cytosine (C), and Thymine (T) at specific nucleotide positions within a locus [67]. The SNP markers being the most common polymorphism in any organism offer several advantages. They can be automated and uncover hidden polymorphisms not detectable by other markers and methods, thus emerging as a popular marker in molecular marker development [66,67]. Genetic markers have been used in previous studies to clarify the genetic diversity of *Cannabis* species and to identify traits such as sexual phenotypes and chemotype-determining variables [68]. In *Cannabis* research, DNA sequences and molecular markers are preferred over morphological markers for determining genetic diversity; morphological markers can be significantly influenced by environmental factors as noted by Nadeem et al. [69]. Sawler et al. [70] analysed 81 marijuana and 41 hemp cultivars from various environments using 14,031 SNPs identified by GBS protocol. The GBS technique is essential for determining the genetic structure of an individual or a population and plays a significant role in understanding the genetic diversity and breeding of *Cannabis* [70].

The SNP analysis found significant genome-wide differences between hemp and marijuana [70]. In a comparative study, 491,341 SNP markers were identified using whole genome sequencing (WGS) and 23,894 using GBS to assess the genetic diversity, phylogenetic relationship, and population structure of 195 *Cannabis* samples [9]. The findings indicated that hemp varieties clustered separately from marijuana varieties and also highlighted higher levels of heterozygosity in marijuana compared to hemp varieties [9]. Cascini et al. [71] conducted a study on Italian *Cannabis* varieties, identifying SNPs that distinguish between *Cannabis* and hemp. Additionally, the study reported on four functional SNPs likely to induce decreased THCAS activity in hemp plants [71]. Soorni et al. [52] identified 24,710 SNPs across all samples and 29,647 SNPs for 68 Iranian individuals and one from Afghanistan. The majority of SNPs detected in this research were transitions (A/G or C/T), accounting for 62.7% of events, while transversion events (A/C, A/T, C/G, or G/T) accounted for 37.3%. These studies collectively demonstrate the importance of SNP analysis in understanding the genetic diversity and characteristics of *Cannabi*s species.

The genetic relationship between different *Cannabis* strains, particularly between marijuana and hemp varieties, has been a subject of research interest. For example, the *C. sativa* subsp. *indica* marijuana strain from Pakistan was found to be genetically more similar to hemp, rather than other marijuana strains [52]. Similarly, hemp was observed to be closely related to marijuana strains [71]. In contrast, Sawler et al. [70] reported that hemp shares a greater proportion of alleles with *C. sativa* subsp. *indica* than with *C. sativa* subsp. *sativa*, indicating complex genetic differences between marijuana and hemp distributed across the genome, extending beyond loci involved in cannabinoid production.

In *Cannabis* research, there is a lack of comprehensive SNP bead chips compared to other crops [72]. The SNP bead chips need to be developed and tailored specifically for *Cannabis* and are necessary due to genetic separation among *Cannabis* populations and the diverse environments they inhabit. Eurofins [73] addresses this need by developing the *Cannabis* SNP chip called CannSNP90 in 2020. This SNP was created from the genomes of 40 *Cannabis* and hemp plants, representing different types and genders, with whole genome sequencing coverage exceeding 50X using an Illumina NovaSeq technology [73]. In a 2023 study by Garfinkel et al. [72], the researchers used the *Cannabis*-specific SNP array to map 117 loci controlling the day-neutral trait in a segregating F2 population. This study stands out as one of the few to employ the *Cannabis* SNP chip in peer-reviewed research, highlighting the novelty and potential of this technology in *Cannabis* genetics research.

Genomics research has attracted a lot of attention in South Africa in recent years. Advanced technologies, like next-generation sequencing (NGS) and Sanger sequencing, are now widely available in both private and academic South African institutions, supporting this rapidly expanding field. These innovations have completely changed the field of genomic research by making it possible to analyse genetic material in-depth at a speed and accuracy never before possible. To the best of our knowledge, no research has tested for genetic variation in South Africa using the CannSNP90 *Cannabis* chip or any other SNP chip, and not much is known about markers associated with *Cannabis* genetic diversity or about important traits such as cannabinoid expression. Sanger sequencing is the first widely used sequencing method, developed in the 1970s and 1980s [74,75]. It repeatedly creates a new DNA strand using a segment of DNA as a template, utilising DNA polymerase and PCR reaction [69]. On the other hand, next-generation sequencing (NGS) is a major advancement. This method can create several hundred million to several hundred billion DNA bases in a single run and uses the successive insertion of nucleotides into spatially assorted DNA templates [69]. These technologies can be used in whole-genome sequencing to examine the number of SNPs and to take into account the South African varieties present within the *Cannabis* species, as well as in genome-wide association studies (GWAS) and the building of linkage and haplotype maps. Further research and development of these technologies in the *Cannabis* industry in the country could provide significant new understandings of human health, biodiversity, and the genetic basis of life.

## 4. Chemical Profiling of Cannabidiol and Δ^9^-Tetrahydrocannabinol

*Cannabis* is a complex plant with secondary metabolites such as terpenes, terpenophenolics, flavonoids, and cannabinoids [9,76]. There are approximately 200 unique cannabinoids (fatty compounds) found in the plant [3]. The term cannabinoids refers to any of the usual C21 groups of compounds found in *Cannabis* [56]. They are produced and stored in glandular trichomes [77], (Figure 4), which are present in the majority of the plant’s aerial surfaces particularly on the bract and flowers of female plants or in the leaves and stems of both male and female plants [28]. Female plants display a higher abundance of glandular trichomes in their flowers, bracts, and other plant parts than male plants [77]. Abebe [78] found that cannabinoids in plant parts are found in higher concentrations in the following order: bracts (14–25%), flowering tops (5–10%), small leaves (3–7%), and large leaves (1–3%) seeds of the plant.

The cannabinoids of *Cannabis* are synthesized from fatty acids and isoprenoid precursors [67]. The biosynthesis is initiated by two distinct metabolic pathways: the polyketide pathway and the plastidal 2-C-methyl-D-erythritol4-phosphate (MEP) pathway (Figure 5) [28,67]. Cannabigerolic Acid (CBGA) is believed to be a key precursor of Cannabinoid biosynthesis (Figure 5). CBGA is produced by the aromatic prenyltransferase cannabigerolic acid synthase (CBGAS) from olivetolic acid (OA) and geranyl pyrophosphate (GPP) as substrates [28]. It used to produce the non-psychotropic metabolite, tetrahydrocannabinolic acid (THCA) [79], cannabidiolic acid (CBDA), and cannabichromenic acid (CBCA) through the cyclization of a prenyl moiety and with the assistance of tetrahydrocannabinolic acid synthase (THCAS), cannabidiolic acid synthase (CBDAS), and cannabichromenic acid synthase (CBCAS), respectively [23,28,67]. The major cannabinoids identified in *Cannabis* are Δ^9^-THC and CBD [9,79] and are synthesized when the acid forms are exposed to light or heat [28].

### 4.1. Cannabinoids Profiles

The *Cannabis* plant has at least 200 cannabinoid compounds; however, the cannabinoids of interest are Δ^9^-THC and CBD as well as their corresponding acids, THCA and CBDA [79,80]. The compounds can be distinguished using a variety of chromatographic techniques, including High-Performance Liquid Chromatography (HPLC), Thin-Layer Chromatography (TLC), and Gas Chromatography (GC) combined with mass spectrometry [67]. Carvalho et al. [81] conducted the first study to illustrate the chemical composition and potency of medicinal *Cannabis* cultivated in Brazil. The majority of the *Cannabis* samples contained high CBD (CBD total = 3.7%), with a maximum Δ^9^-THC content of 2.6%. Fernández et al. [82] used GC/MS to determine Δ^9^-THC, CBD, and Cannabinol in *Cannabis* samples from Argentina. There were different cannabinoids determined, namely, Δ^9^-THC, CBD, Cannabinol (CBN), cannabichromene (CBC), cannabigerol (CBG), THCA, and CBDA [82]. To account for variations in cannabinoid concentrations, Galettis et al. [80] developed various ranges of calibration curves. The ranges for THCA and CBDA were 1–150 mg mL^−1^; Δ^9^-THC and CBD were 0.5–75 mg mL^−1^; and the remaining cannabinoids were 0.5–20 mg mL^−1^.

A recent study by Wishart et al. [83] used LC-MS/MS to detect and quantify 16 cannabinoids, and THCA was the most abundant compound found in the six tested *Cannabis* cultivars (concentration range 133 mg/g–162 mg/g), while cannabidivarin (CBDV) was the least abundant compound (concentration range 0.312 µg/g to 1.58 µg/g). Mthembi et al. [84], used GC-MS on a South African drug called nyaope. Nyaope is a cocktail mixture of low-grade heroin smoked with dried *Cannabis* [84]. The authors were able to determine 16 compounds; among those compounds, nine cannabinoids were firmly identified: cannabivarin (CBV), cannabicyclol (CBL), Δ^9^-THC, CBD, CBC, cannabicoumoronone (CBCN), tetrahydrocannabivarin (THCV), CBG, and CBN [84]. This suggests that South African *Cannabis* varieties are particularly rich in these compounds. Similarly, other reports have investigated various South African *Cannabis* strains or landraces and their cannabinoid profiles [85]. It was found that Mpondo Gold has a high Δ^9^-THC concentration (about 20%) and an earthy and woody *Cannabis* aroma [85,86], whereas the *Cannabis* strain called Durban Poison is known for its sweet, fruity flavour, high Δ^9^-THC level ranging from 15% and 25%, low CBD level (0.1–0.3%), and CBG (0.6–1.4%) and THCV (0.2–1.8%) [81]. The Eswatini Swazi Gold produces dense mango-smelling buds and has a high Δ^9^-THC content that ranges from 18% to 27%, and high THCV (1–3%) [14,86]. KwaZulu or IsiZulu is a pure *sativa* landrace strain that originated from the slopes of Drakensberg ridge in South Africa and has a Δ^9^-THC content above 20% [86]. Research in *Cannabis* has shown that African strains have various varieties that are rich in the identified Δ^9^-THC homologue known as THCV [85]. With detailed knowledge of cannabinoid profiles, local breeders can engage in more informed breeding programs aimed at enhancing desirable traits such as higher yields of particular cannabinoids, resistance to local pests and diseases, or better adaptability to the South African climate. This may result in the development of exclusive strains that the local industry will benefit greatly from by being able to sell and patent.

The South African Health Products Regulatory Authority (SAPHRA) regulates the cultivation of *Cannabis* solely for medicinal, scientific, and clinical purposes through permitted licenses [2]. This initiative has opened the door for this plant species to be legally researched for medical, industrial, and recreational purposes. Research on chemical profiling in local *Cannabis* varieties will help in identifying the unique chemical composition such as cannabinoids and terpenes and also their concentrations. This is crucial for medicinal and clinical research as well as further assisting in selecting the varieties with interesting chemical profiles to facilitate breeding of the plant. The SAPHRA regulations limit the amounts of CBD (0.0075%) and Δ^9^-THC (0.001%) in *Cannabis* strains grown for medicinal purposes, distribution, and sale of *Cannabis* products [2]. The use of Chromatographic techniques can assist in the analysis of these concentrations of cannabinoids. In South Africa, these techniques aid in the local varieties’ cannabinoids and terpene identification and quantification, supporting research, regulatory compliance, and quality control. Further assisting in educating the local *Cannabis* farmers regarding the correct amounts of CBD and THC needed to be grown according to SAPHRA. This will contribute to understanding the unique properties of these strains and their potential benefits, contributing to both scientific knowledge and the development of new therapeutics. Chromatography will play an important role in *Cannabis* analysis as the legal landscape develops, guaranteeing the plant’s safe and efficient use.

### 4.2. Cannabinoids Diversity and Genomics

*Cannabis* has different chemical variants showing chemical diversity and morphological variations known as chemotypes [87,88]. *Cannabis* is divided into four chemotypes based on the Δ^9^-THC: CBD ratio. The Δ^9^-THC: CBD ratio is used to distinguish between high and low THC-containing plants [87]. The diversity in the chemotype of *Cannabis* can be further supported by using DNA markers with genes coding for Tetrahydrocannabinolic Acid Synthase (THCAS) and cannabidiolic acid synthase (CBDAS) [23]. In the Australian *Cannabis* study, Hurgobin et al. [29] developed a Mendelian inheritance model of chemotype. This model comprised a single locus B, with two co-dominant alleles, *B_T_* and *B_D_*, encoding for THCAS and CBDAS, respectively. The homozygosity at the B locus produces either Δ^9^-THC (chemotype I (drug-type), *B_T_*/*B_T_*) or CBD (chemotype III (fibre-type), *B_D_*/*B_D_*). While heterozygous individuals (*B_T_*/*B_D_*) produce a mixed THC-CBD belonging to chemotype II [29]. Garfinkel et al. [72] reported that chemotype IV contains the null allele *B_O_*, which is composed of CBGA, the precursor compound of both CBDA and THCA. The *B_O_* allele is biochemically unable to convert CBGA into THCA or CBDA, thus leading to CBGA accumulation in chemotype IV [72].

*Cannabis* genomics research has focused mostly on the characterization of genes underlying the production of cannabinoids (THC and CBD) [89]. According to Sawler et al. [70], SNPs were used to identify marijuana and hemp in Canadian *Cannabis*. The hemp was genetically closer to *C. sativa* subsp. *indica* than to the *C. sativa* subsp. *sativa*. Marijuana and hemp have differing capacities for cannabinoid biosynthesis, with marijuana having the *B_T_* gene for THCAS and hemp having the *B_D_* allele for CBDAS [70]. Studies suggest that the distinction between these populations is not limited to genes underlying THC synthesis [70].

In a study by Singh et al. [23], Canadian *Cannabis* showed multiple linked loci with alleles at various loci that have been identified in genomic investigations. The THCA/CBDA variation is thought to be due to sequence variations at the *B_T_* and *B_D_* loci [23]. In the Canadian stains, the D589 marker determined the presence of solely active THCAS (the *B_T_* allele) but the B1080/B1192 marker determined the presence of both THCAS and CBDAS synthases (*B_T_* and *B_D_* alleles, respectively) [23].

Divergence at the CBDAS loci on the other hand is primarily responsible for setting the THCA: CBDA ratio of cultivars resulting in differences in cannabinoid profiles between marijuana and hemp [24]. The authors also reported that variation in THCAS and CBDAS copy numbers had also contributed to a wide range of cannabinoid content in *Cannabis* strains, as well as phytochemical variety, which contributes to plant adaptation [24]. Variations in *Cannabis* chemotypes are influenced by both environmental and genetic factors [23,70]. Therefore, it is critical to comprehend how the genetics of *Cannabis* grown in various geographic locations relate to the synthesis of significant cannabinoids like THC and CBD. As previously stated, there was no published research on *Cannabis* genetics or cannabinoid profiling in the southern African region. Furthermore, there are no published South African or African representative genomes in the 15 genomes released by the NCBI as mentioned before. At present, there is no proper identification of *Cannabis* chemotypes throughout the continent. To successfully breed *Cannabis* cultivars with the appropriate level and ratio of cannabinoids for pharmaceutical applications as well as for breeding strategies, a thorough screening of the germplasm collection is required [23]. Chromatographic techniques such as HPLC and GC offer an advanced way to quantify the cannabinoid and signatures (chemical fingerprints), a tool for identifying the place of origin and cultivation of the *Cannabis* plant [67,78]. This research could be further supported by the use of molecular markers such as SNPs associated with gene coding for THCAS and CBDAS [23]. These will reveal the relationship between cannabinoid profiles and genetic variations for plant breeding, hemp cultivation, and the medical and pharmaceutical industry in South Africa. As a result, this will directly benefit the nation’s social and economic landscape by generating employment and a vigorous *Cannabis* industry.

## 5. Conclusions

The landscape of *Cannabis* legislation has undergone significant changes in recent years, prompting increased interest in the plant across various sectors [2]. Researchers, policymakers, entrepreneurs, and the public have all shown heightened interest in *Cannabis* due to evolving legal frameworks. This surge in interest has prompted a wealth of literature exploring various aspects of *Cannabi*s, including its genetics, chemical profile, and potential applications. *Cannabis,* known as the oldest crop, has been cultivated for between 3000 and 6000 years [8]. During the 1500s and 1700s, European and Portuguese settlers introduced the crop to South Africa through trade and travel, where it has been grown for many years, reflecting the plant’s historical significance [8,12]. Moreover, South Africa may harbour a rich genetic reservoir for this species that remains unidentified. Despite the existence of 15 genome assemblies available from the National Centre for Biotechnology Information (NCBI), none of them are representative of the genomes of South Africa or Africa. Additionally, while the *Cannabis* plant contains over 200 distinct compounds [3], these compounds are not properly identified in South Africa. Further research is required to fully comprehend the genetic diversity and phytochemical profiles found in South African *Cannabis*. This manuscript has shed light on numerous gaps in our understating of *Cannabis* genetics and chemical profiling by addressing these gaps through research and literature. Stakeholders can gain valuable insights into *Cannabis* genetics, cultivation, and utilization, thereby facilitating advancements across various sectors and the development of more effective policies and practices. Creating a reference genome for local varieties can result in major advancements in science, agriculture, and conservation by allocating resources. These advancements will eventually benefit the local community and advance global knowledge. Chromatographic methods are essential for the detection and measurement of chemical profiles in South African local *Cannabis* landraces. Their applications in industry ensure product quality, safety, and regulatory compliance, while in academic institutions, they support extensive research and education. Chromatography will continue to play a critical role in these fields as the *Cannabis* industry and scientific interest in its properties grow.

## Figures and Tables

**Figure 1 plants-13-02695-f001:**
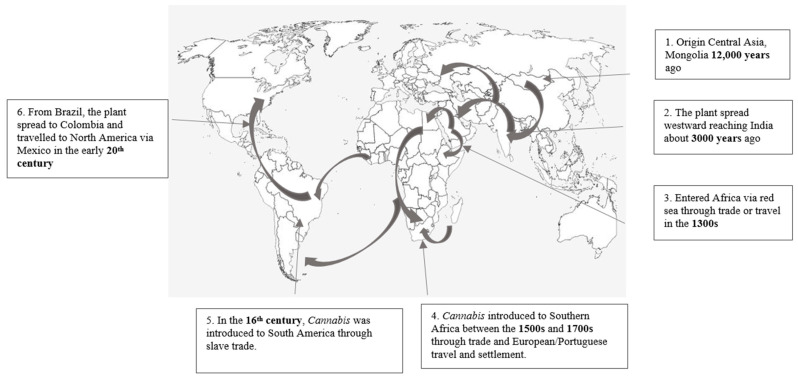
The brief history and movement of *Cannabis* (illustration adapted from [4,10,12]).

**Figure 2 plants-13-02695-f002:**
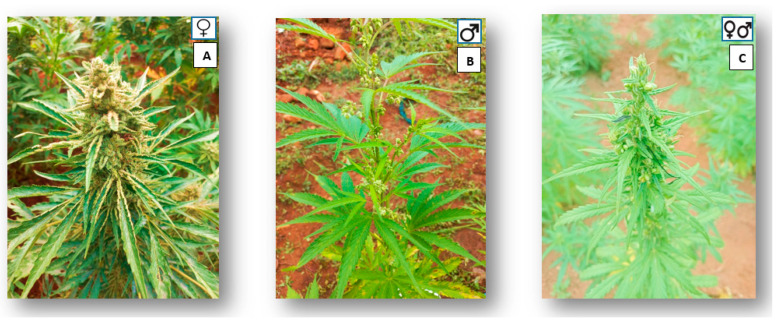
Morphological differences between 2 (**A**) female, 2 (**B**) male plants, and 2 (**C**) hermaphrodite plants of *Cannabis* (illustration by V.V Ndlangamandla).

**Figure 3 plants-13-02695-f003:**
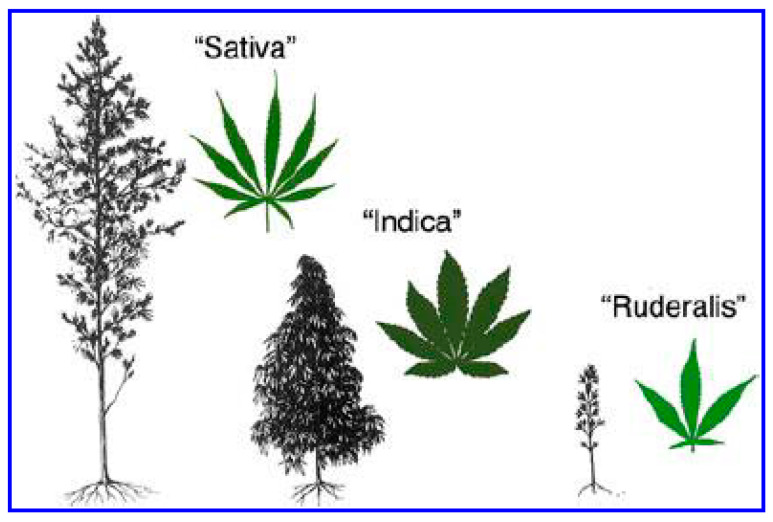
The morphological variation of *C. sativa* subsp. *sativa*, *C. sativa.* subsp. *Indica,* and *C. sativa* subsp*. ruderalis* (illustration from [55]).

**Figure 4 plants-13-02695-f004:**
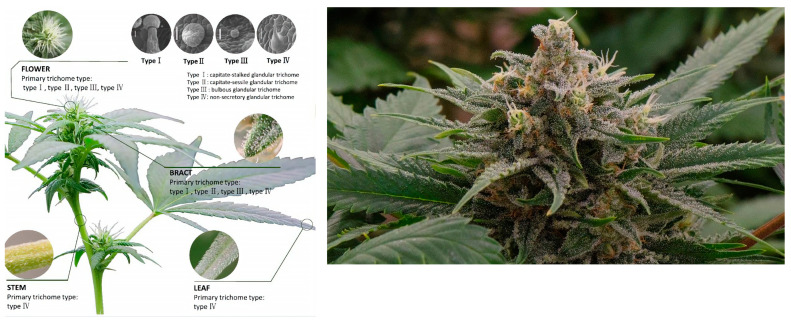
Glandular trichomes on the surface of the female flower, stem, and leaf of *Cannabis*, adapted from [77].

**Figure 5 plants-13-02695-f005:**
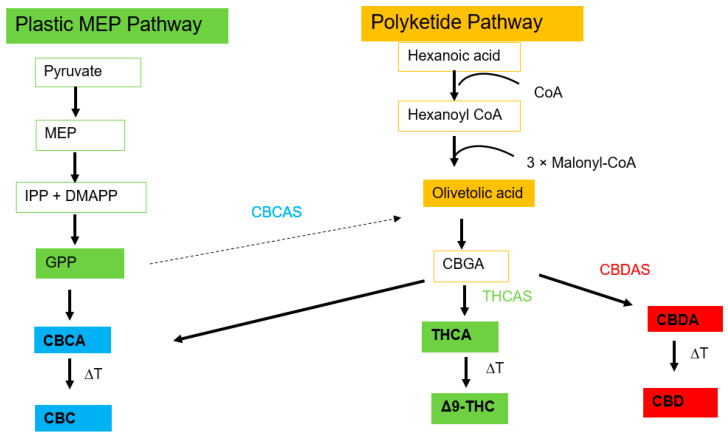
The biosynthesis pathway leading to the synthesis of cannabinoids, the adapted form from the previous studies [28,67].

**Table 1 plants-13-02695-t001:** Medicinal uses of *Cannabis sativa* in most relevant studies.

Botanical Name	Medicinal Application	Product or Plant Part	Country	References
1. *C. sativa* L.	To treat bacterial diseases such as cholera, tetanus, and chronic pain.	Tincture of hemp and hemp resin	England	[31]
2. *C. sativa* L.	Appetite stimulation, nausea, vomiting and pain relief	Dronabinol	Canada	[33]
3. *C. sativa* L.	Anti-inflammatory properties- in lung inflammation in cases of COVID-19	CBD *Cannabis* extract	Poland	[30]
4. *C. sativa* L.	Chronic non-cancer pain, Insomnia and anxiety	CBD *Cannabis* extract	Australia	[35]
5. *C. sativa* L.	Epilepsy, nausea/vomiting, and diarrhoea associated with the use of chemotherapy	Sativex^®^, Epidiolex^®^, and Dronabinol	Australia	[34]
6. *C. sativa* L.	Pain relief for neuropathic pain in adult patients with sclerosis and late-stage cancer	Sativex^®^	Canada	[33]
7. *C. sativa* L.	Treatment of AIDS-related anorexia and nausea and vomiting associated with chemotherapy	Dronabinol	Italy	[36]
8. *C. sativa* L.	Nausea and vomiting associated with chemotherapy and unresponsive to conventional therapies	Nabilone	Germany	[37]
9. *C. sativa* L.	To treat convulsions, Dravet syndrome and Lennox–Gastaut syndrome, algesic therapy in cancer patients with severe pain	Herbal *Cannabis,* Epidiolex^®^, and Sativex^®^	Israel	[38]
10. *C. sativa* L.	Used in Crohn’s disease, ulcerative colitis, itching, migraine, post-traumatic stress disorders and Alzheimer’s disease	Bedrocan^®^	Netherlands	[34]
11. *C. sativa* L.	To treat inflammatory intestinal disturbances and irritable intestine syndrome	*Cannabis* extract	United States of America	[34]

**Table 2 plants-13-02695-t002:** Traditional medicine uses of *Cannabis sativa* in South Africa.

Traditional Uses	Plant Part	Origin	References
To treat hypertension (high blood pressure), ulcers, chest complaint, asthma and shortness of breath, promote weight loss, poisoning by ingestion (isidliso)	Leaves	KwaZulu-Natal	[43]
To treat diabetes, hypertension, obesity, and stroke	Leaves	KwaZulu-Natal	[42]
Used to treat impotency, diarrhoea, indigestion, epilepsy, insanity, and chronic diabetes	Leaves and stems	Limpopo	[45]
Used as a cure for asthma, bronchitis, headache, flu, epilepsy, cough, and pains	Leaves and flowers	Free State	[44]

## Data Availability

Not applicable.

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
