# Peer review of "Finally Freed—Cannabis in South Africa: A Review Contextualised within Global History, Diversity, and Chemical Profiles"

_plants, 2024, doi:10.3390/plants13192695_

Round 1

Reviewer 1 Report (Previous Reviewer 1)

Comments and Suggestions for Authors

- Page 12/ Line 451: Find: to induce decreased THCAS activity in hemp plants 

Replaced with: to induce decreased tetrahydrocannabinolic acid synthase (THCAS) activity in hemp plants

- Page 16/ line 600: Find: further supported by using DNA markers with genes coding for Tetrahydrocannabinolic 600 Acid Synthase (THCAS) and cannabidiolic acid synthase (CBDAS)

Replaced with: further supported by using DNA markers with genes coding for THCAS and cannabidiolic acid synthase (CBDAS)

Page 6/ Table 1: Delete the column of Botanical name (C. sativa L).

Page 6/ Table 1: Find the title of Table 1: Medicinal uses of Cannabis in most relevant studies

Replaced with: Medicinal uses of Cannabis sativa L in most relevant studies

Check and use the same formate of all subtitles

e.g.

2.1.2 The summary of the South African bill of Cannabis for private purposes

2.1.1 Decriminalisation of the Use of Cannabis in South Africa

3.1 Botanical Taxonomy of Cannabis

3.2 Cannabis morphological diversity

3.4 Population structure of Cannabis using Single nucleotide polymorphisms (SNPs)

-Page 12/Line 425: Find: Single nucleotide polymorphisms (SNPs) have become increasingly significant

Replaced with: SNPs have become increasingly significant

- Insert the project number of funding.

- Check the formate of all references e.g. Ref. No. 65, 67, 79, 80, 89, 92

Author Response

Comments 1:

- Page 12/ Line 451: Find: to induce decreased THCAS activity in hemp plants 

Replaced with: to induce decreased tetrahydrocannabinolic acid synthase (THCAS) activity in hemp plants

- Page 16/ line 600: Find: further supported by using DNA markers with genes coding for Tetrahydrocannabinolic 600 Acid Synthase (THCAS) and cannabidiolic acid synthase (CBDAS)

Replaced with: further supported by using DNA markers with genes coding for THCAS and cannabidiolic acid synthase (CBDAS)

Response 1: Agreed.

- The term THCAS has been written in full and abbreviated in brackets, tetrahydrocannabinolic acid synthase (THCAS) (line 452). 

- tetrahydrocannabinolic acid synthase was abbreviated (find previous paper, lines 522-523 and current paper, line 523).

- tetrahydrocannabinolic acid synthase and cannabidiolic acid synthase were abbreviated (find line 600).

Comments 2: - Page 6/ Table 1: Delete the column of Botanical name (C. sativa L).

Response 2: Agreed. The column of Botanical name (C.sativa L) was deleted (find lines 186- 187).

Comments 3:

- Page 6/ Table 1: Find the title of Table 1: Medicinal uses of Cannabis in most relevant studies

-Replaced with: Medicinal uses of Cannabis sativa L in most relevant studies

Response 3: The title of Table 1 has been updated to "Medicinal uses of Cannabis sativa L in most relevant studies" (line 185 )

Comments 4:

 - Check and use the same formate of all subtitles

e.g.

2.1.2 The summary of the South African bill of Cannabis for private purposes

2.1.1 Decriminalisation of the Use of Cannabis in South Africa

3.1 Botanical Taxonomy of Cannabis

3.2 Cannabis morphological diversity

3.4 Population structure of Cannabis using Single nucleotide polymorphisms (SNPs)

Response 4: Agreed. We have reviewed and updated the subtitles to ensure they follow a consistent format.

- We have changed the subtitle “2. Origin and History of Cannabis” to “2. Origin, History and Uses of Cannabis” (line 63).

- Standardized the capitalization style to title case for uniformity.

- Applied a consistent hierarchical numbering system for all subtitles.

Comments 5: -Page 12/Line 425: Find: Single nucleotide polymorphisms (SNPs) have become increasingly significant

Replaced with: SNPs have become increasingly significant

Response 5: Single nucleotide polymorphisms was abbreviated (previous paper, line 425; current paper, line 426).

Comments 6: - Insert the project number of funding.

Response 6:

The project number was added (current paper, line 685) 

Comments 7: - Check the formate of all references e.g. Ref. No. 65, 67, 79, 80, 89, 92

Response 7: Agreed. All references were verified and checked.

-        Changes were made on ref No. 3, 8, 9, 11, 19, 29, 60, 63, 65, 67, 79, 80, 89, and 92.

Reviewer 2 Report (Previous Reviewer 2)

Comments and Suggestions for Authors

In my view, the current version of the manuscript is now ready to be published in this journal. Congratulation to authors for such a comprehensive study. 

Author Response

Comments 1: In my view, the current version of the manuscript is now ready to be published in this journal. Congratulation to authors for such a comprehensive study.

Response 1: Thank you for your positive feedback and for recognizing the effort put into this review. We are pleased to hear that you find the manuscript ready for publication. We appreciate the opportunity to contribute to this journal and are grateful for your guidance throughout the review process.

Reviewer 3 Report (Previous Reviewer 3)

Comments and Suggestions for Authors

The modifications to the initial submission are appreciated and help clarify the aim and scope of the paper from one focused on detailing the state of the art, to one focusing more on the potential research focused on South African cultivars of Cannabis can have on the economic livelihood of its citizens.  However since the discussion is focused more on potential the title itself is rather misleading.  If there is nothing to report on the specifics of SA cannabis, its diversity and potential in any number of commercial endeavors, it reduces to a review of cannabis and its history and chemical composition.  My sense is that this subject has been more adequately covered elsewhere.  One recommendation is that the authors re-structure this paper more as a policy position to support future investment into this area of research.  As such it might fit more neatly in the journal Agriculture that has in its aims and scope a section on agricultural development.

Author Response

Comments 1:  The modifications to the initial submission are appreciated and help clarify the aim and scope of the paper from one focused on detailing the state of the art, to one focusing more on the potential research focused on South African cultivars of Cannabis can have on the economic livelihood of its citizens.  However since the discussion is focused more on potential the title itself is rather misleading.  If there is nothing to report on the specifics of SA cannabis, its diversity and potential in any number of commercial endeavors, it reduces to a review of cannabis and its history and chemical composition.  My sense is that this subject has been more adequately covered elsewhere.  One recommendation is that the authors re-structure this paper more as a policy position to support future investment into this area of research.  As such it might fit more neatly in the journal Agriculture that has in its aims and scope a section on agricultural development.

Response 1:

Thank you for your detailed and thoughtful feedback on our manuscript. We appreciate your recognition of the modifications made to clarify the aim and scope of our paper, shifting the focus towards the potential research on South African Cannabis cultivars and their economic impact. We feel that our submission fits within the scope of the Plant Genomics, Genetics and Biotechnology section. Our paper aims to describe the application of genomic technologies and biotechnology to Cannabis research and emphasize the research gaps that South Africa currently has on Cannabis, aligning with the journal's focus areas. We also want to note that the two other reviewers have perceived the review as within the scope of novel contributions to the field. In addition, the suggestion that the paper as a policy document will be considered once we have completed the detailed study on indigenous practices and diversity of the landraces in SA. We note and appreciate your comment on the title, however it would be appreciated if you could suggest a more appropriate title for the manuscript.

This manuscript is a resubmission of an earlier submission. The following is a list of the peer review reports and author responses from that submission.

Round 1

Reviewer 1 Report

Comments and Suggestions for Authors

Dear:
Thank you for submitting the manuscript.

Comments and suggestions

-          Check the use of term cannabidiol acid or cannabidiolic acid. Example L 478 and L 485

Another example: Check the use of term Δ9-tetrahydrocannabinol  or tetrahydrocannabinol.  Example L 164, L 437, L 466, L 477, and  L478

Another example: L 484: tetrahydrocannabinolic acid   (THCA) or L 478: tetrahydrocannabinol acid (THCA)

- Abbreviations and acronyms stand for when they are first used. For example

-Cannabidiol L 164, L 437, L 466, 477

-cannabidiol acid L 478, L 484

-L 164: age of Δ9-tetrahydrocannabinol (THC)

L478 : tetrahydrocannabinol acid (THCA) and cannabidiol acid

tetrahydrocannabinol (THC) and Cannabidiol (CBD)

Example:-L 462: CBGA synthase (CBGAS) using olivetolic acid…and

Example: L 464: cannabichromeric acid synthase (CBCAS) are …

- L 210: It is recommended to delete Table 2: Traditional medicine uses of Cannabis in South Africa. The table includes one species (Cannabis sativa).

-          L 472: Find: Figure 5. The biosynthesis pathway leading to synthesis of cannabinoids, adapted from [32,65].  Replaced with Figure 5. The biosynthesis pathway leading to synthesis of cannabinoids, the adapted form from the previous studies [32,65]

-           

-          Check the format of all references e.g.  No. 31-27, 64 and 71 -74.

-          -Check the presence of DOI

-          Insert briefly the impact of the following references:

Cannabis for Medical Use-A Scientific Review. 2017,

 International perspectives on the implications of cannabis legalization: A systematic review & thematic analysis. Int J Environ Res Public Health 2019, 16, 3095

and  the review of: Extraction  of cannabinoids from Cannabis sativa L. (hemp)-review. Agriculture 2021, 11.

 - L475 –L 504: Insert recent references especially in the section of 4.1 Cannabinoids profiles. Example

-Wishart DS, Hiebert-Giesbrecht M, Inchehborouni G, Cao X, Guo AC, LeVatte MA, Torres-Calzada C, Gautam V, Johnson M, Liigand J, Wang F. Chemical composition of commercial cannabis. Journal of agricultural and food chemistry. 2024 Jan 5.

-Wong-Salgado P, Soares F, Moya-Salazar J, Ramírez-Méndez JF, Moya-Salazar MM, Apesteguía A, Castro A. Therapeutic Potential of Cannabinoid Profiles Identified in Cannabis L. Crops in Peru. Biomedicines. 2024 Jan 29;12(2):306.

-O'Brien, Michael. "cannabinoids". Encyclopedia Britannica, 21 Mar. 2024, https://www.britannica.com/science/cannabinoid. Accessed 5 April 2024.

- Mthembi PM, Mwenesongole EM, Cole MD. Chemical profiling of the street cocktail drug ‘nyaope’in South Africa using GC–MS I: Stability studies of components of ‘nyaope’in organic solvents. Forensic science international. 2018 Nov 1;292:115-24.

Comments on the Quality of English Language

Minor editing of English language required.

Reviewer 2 Report

Comments and Suggestions for Authors

This is a very short and sound review of our most recent knowledge about the Cannabis sativa plant. It is well-organised ans rich in latest bibliographical items cited. I'd suggest the authors to consider only those minor issues I have found:

In lines 296-297 the Authors stated: "other scientists have divided Cannabis into three distinct 296 species."

The interpretation of this fact is different: As a result of continuous taxonomic research, within the genus Cannabis L. a considerable number of names have been proposed at the rank of species or intraspecific taxa, and their total number reached 296. The reason for such an excess of names is the great morphological habitat-dependent variability of cannabis, to which the botanists tried to assign a taxonomic value. All these historical names have today lost their taxonomic value in favor of a single species, C. sativa L. Only some authors are inclined to recognize three taxa: C. sativa, C. indica and C. ruderalis. 

lines 297-304

Please add the authorship to the binomials mentioned, especially to the Cannabis indica and Cannabis ruderalis

line 540

has also contributed -> had also contributed

line 545

there is no -> there was no

In my view, the manuscript deserves to be published in your journal.

End of review.

Reviewer 3 Report

Comments and Suggestions for Authors

This was a challenging paper to review.  It’s title “Finally Freed - Cannabis in Southern Africa: A review of the history, diversity and chemical profile” certainly implies that the reader will hear something about the history, diversity and chemical profiles of South African Cannabis, however it seems to misrepresent the scope of the paper.  Instead, the authors have developed a wide-ranging review of the species that draws from literature that has itself already been reviewed including the species taxonomy, its ethnobotanical history of dispersal around the world, any medical research involving its secondary chemistry and any genomic resources that have been developed for Cannabis.  This background information is largely derived from literature outside of South Africa.  Most of the direct mentions of Southern Africa are generic --like the new legal and regulatory framework in the country, it’s current and potential economic impacts and a general, anecdotal description of distinctive landrace varieties grown in Southern Africa. 

This is useful information however it really has little to offer in terms of the history, genetic diversity and chemical profiles within this region. Interestingly, what is directly emphasized is the lack of knowledge in this species in this part of the world: no specific genomic reference (ln 380), no markers (ln 420), no high-throughput SNP assays (ln 434), no peer reviewed chemical profiling of local varieties (503) and no effort yet to develop reference genomes for South African landraces (ln 546). The title is is more aspirational than what the text provides.  The gaps that are mentioned are not addressed more than mentioning them.  The authors don’t follow up on these research deficiencies with broader questions about how to develop in the future given this new transition to leagalization. What would a comprehensive assessment look like?  What would be the prioritization and resources needed to develop a reference genome of local varieties and how could the genomic data provide more insight into the identity and diversity of varieties grown in the region?  What kinds of chemical profiles might be developed in partnership with both industry and academic institutions labs already doing these kinds of assays?  How would this help support more opportunities for the Cannabis industry in this region?

My sense it the authors have developed a persuasive grant proposal.  The scope and rationale sections provide enough background and the gaps identified provide justification for future funding.

Comments on the Quality of English Language

The english is fine